# Rhodium-Catalyzed *Trans*-Bis-Silylation Reactions of 2-Ethynyl-3-pentamethyldisilanylpyridines

**DOI:** 10.3390/molecules28083284

**Published:** 2023-04-07

**Authors:** Akinobu Naka, Hisayoshi Kobayashi

**Affiliations:** 1Department of Life Science, Kurashiki University of Science and the Arts, Nishinoura, Tsurajima-cho, Kurashiki 712-8505, Okayama, Japan; 2Emeritus Kyoto Institute of Technology, Matsugasaki, Kyoto-shi 606-8585, Kyoto, Japan

**Keywords:** rhodium-catalyzed reaction, *trans*-bis-silylation, silole, DFT calculation, 2-ethynyl-3-pentamethyldisilanylpyridine, cyclic silicon compound, Sonogashira coupling reaction, palladium-catalyzed reaction, copper-catalyzed reaction, energy diagram

## Abstract

Rhodium-catalyzed reactions of 2-ethynyl-3-pentamethyldisilanylpyridine derivatives (**1** and **2**) are reported. The reactions of compounds **1** and **2** in the presence of catalytic amounts of rhodium complexes at 110 °C gave the corresponding pyridine-fused siloles (**3**) and (**4**) through intramolecular *trans*-bis-silylation cyclization. The reaction of 2-bromo-3-(1,1,2,2,2-pentamethyldisilanyl)pyridine with 3-phenyl-1-propyne in the presence of PdCl_2_(PPh_3_)_2_-CuI catalysts afforded 1:2 bis-silylation adduct **6**. DFT calculations were also performed to understand the reaction mechanism for the production of compound **3** from compound **1**.

## 1. Introduction

Various synthetic methods of organosilicon compounds have been reported so far [1]. Silicon-containing compounds, due to their unique physical and chemical properties, are attractive as candidates for optical and electronic materials such as organic thin film transistors, organic light-emitting diodes, and organic photovoltaics [2,3,4,5,6,7]. In particular, the synthesis and properties of silole derivatives with low-lying LUMO have been extensively studied. The bis-silylation of unsaturated carbon compounds, which provides two silicon-carbon bonds simultaneously, has been developed [8,9,10,11,12,13,14,15,16,17,18,19,20,21,22,23]. Bis-silylation reactions using various substrates have been reported so far, but most of them are *cis*-additions of a silicon–silicon bond to unsaturated compounds. This severely limits the practicality of the method. It seems to be very important that *trans*-bis-silylation of alkynes can be carried out easily and successfully. There are several reports on the *trans*-bis-silylation reactions of alkynes. In 2012, Matsuda and co-workers showed the possibility of a *trans*-selective bis-silylation reaction of the C–C triple bond in the Rh(I)-catalyzed intramolecular cyclization of specific (2-alkynylphenyl)disilanes [18]. The reaction mechanism of the rhodium-catalyzed reactions has not yet been clarified.

Recently, we reported that the reactions of 2-bromo-3-(pentamethyldisilanyl)pyridine with ethynylbenzene derivatives in the presence of PdCl_2_(PPh_3_)_2_-CuI as catalysts afforded the corresponding pyridine-fused siloles through intramolecular *trans*-bis-silylation [24]. DFT calculations for the above reaction were performed to rationalize the formation of *trans*-bis-silylation adducts via *cis*-bis-silylation adducts. We also demonstrated that the similar reactions of 2-bromo-3-(pentamethyldisilanyl)pyridine with alkynes having bulky substituents, such as ethynyltrimethylsilane, produced 2-ethynyl-3-pentamethyldisilanylpyridine derivatives arising from Sonogashira-coupling reactions [24]. Pyridine-containing materials have been examined for their optical and physical properties, as well as their medical potential [25].

It is of considerable interest to us to investigate the chemical behavior of 2-ethynyl-3-pentamethyldisilanylpyridine derivatives in the presence of rhodium catalysts to synthesize pyridine-fused silole derivatives. In this paper, we report the rhodium-catalyzed reactions of 2-ethynyl-3-pentamethyldisilanylpyridine derivatives, and DFT calculations to investigate the energy and structural changes in the synthesis route from 3-(1,1,2,2,2-pentamethyldisilanyl)-2-(trimethylsilylethynyl)pyridine (**1**) to *trans*-bis-silylation product **3**.

## 2. Results and Discussion

### 2.1. Synthesis and Reactions

The starting compound, 3-(1,1,2,2,2-pentamethyldisilanyl)-2-(trimethylsilylethynyl)pyridine (**1**), was prepared by the Sonogashira coupling reaction of 2-bromo-3-(1,1,2,2,2-pentamethyldisilanyl)pyridine with ethynyltrimethylsilane in triethylamine [24]. It was shown that base desilylation can be accomplished in potassium carbonate/methanol to produce 2-ethynyl-3-(1,1,2,2,2-pentamethyldisilanyl)pyridine (**2**) (Figure 1).

We first examined the reaction of compound **1** in the presence of Di-μ-chloro-tetracarbonyldirhodium(I), [RhCl(CO)_2_]_2_. The treatment of compound **1** in the presence of a catalytic amount of [RhCl(CO)_2_]_2_ in refluxing toluene for 12 h gave 1,1-dimethyl-2,3-bis(trimethylsilyl)-1*H*-silolo(3,2-*b*)pyridine (**3**) in 46% yield (Figure 2). Many unidentified products were detected in the reaction mixture by GLC and GPC. Compound **3** was obtained via the intramolecular *trans*-bis-silylation of compound **1**. The structure of compound **3** was verified by spectroscopic analysis. The mass spectrum for compound **3** showed parent ions at *m*/*z* 305, corresponding to the calculated molecular weight of C_15_H_27_NSi_3_. The ^1^H NMR spectrum for compound **3** showed signals at 0.29, 0.34, and 0.38 ppm due to the methyl protons on the silicon atoms, and three doublets of doublet signals at 6.98, 7.73, and 8.46 ppm due to the pyridyl ring protons. The ^29^Si NMR spectrum for compound **3** showed signals at −9.8, −6.6, and 10.1 ppm.

The intramolecular *trans*-bis-silylation of compound **1** proceeded in the presence of [RhCl(nbd)]_2_ (nbd = norborna-2,5-diene) to afford compound **3** in 6% yield. Many unidentified products were detected in the reaction mixture by GLC and GPC. Similar reactions of compound **1** in the presence of RhCl(PPh_3_)_3_ gave compound **3** in 5% yield. The starting compound **1** was recovered (87%) (Table 1).

The reactions of 2-ethynyl-3-(1,1,2,2,2-pentamethyldisilanyl)pyridine (**2**) bearing a terminal alkyne moiety in the presence of a catalytic amount of [RhCl(CO)_2_]_2_ afforded 1,1-dimethyl-3-(trimethylsilyl)-1*H*-silolo(3,2-*b*)pyridine (**4**) in 48% yield. The [RhCl(nbd)]_2_ catalyzed reaction of compound **2** afforded compound **4** in 5% yield. Many unidentified products were detected in the reaction mixture by GLC and GPC. When RhCl(PPh_3_)_3_ was used as the catalyst, compound **2** failed to yield compound **4**. The starting compound **2** was recovered (80%) (Table 1).

We carried out the Sonogashira coupling reaction of 2-bromo-3-(1,1,2,2,2-pentamethyldisilanyl)pyridine with 3-phenyl-1-propyne in triethylamine to obtain 2-(benzylethynyl)-3-(1,1,2,2,2-pentamethyldisilanyl)pyridine (**5**). When a mixture of 2-bromo-3-(1,1,2,2,2-pentamethyldisilanyl)pyridine and 1.9 equivalent of 3-phenyl-1-propyne in the presence of a catalytic amount of PdCl_2_(PPh_3_)_2_-CuI was heated to reflux in triethylamine, intermolecular bis-silylation product **6** produced from the reaction of compound **5** with 3-phenyl-1-propyne was obtained in 23% yield (Figure 3). No compound **5**, which is a 1:1 adduct of 2-bromo-3-(1,1,2,2,2-pentamethyldisilanyl)pyridine and 3-phenyl-1-propyne, was detected by spectroscopic analysis. The similar reaction of 2-bromo-3-(1,1,2,2,2-pentamethyldisilanyl)pyridine with slightly less (0.96 equivalent) 3-phenyl-1-propyne gave compound **6** in 12% yield based on 2-bromo-3-(1,1,2,2,2-pentamethyldisilanyl)pyridine. All attempts to obtain compound **5** were unsuccessful. The ^1^H NMR spectrum for compound **6** revealed two signals at 0.10 and 0.45 ppm due to the methylsilyl protons and a signal at 6.18 ppm attributed to olefinic proton, as well as methylene protons, phenyl, and pyridyl ring protons. The ^13^C NMR spectrum of compound **6** showed two resonances at 26.3 and 37.6 ppm, attributed to methylene carbons, and two resonances at 85.7 and 106.4 ppm due to sp carbons, as well as methylsilyl carbons and phenyl, pyridyl ring, and olefinic carbons. Its ^29^Si NMR spectrum showed two signals at −22.2 and −3.3 ppm.

We carried out the reaction of 2-bromo-3-(1,1,2,2,2-pentamethyldisilanyl)pyridine with 1-hexyne in the presence of a PdCl_2_(PPh_3_)_2_-CuI catalyst. Many products were detected in the reaction mixture by GLC and GPC, and all attempts to isolate 2-(hex-1-yn-1-yl)-3-(1,1,2,2,2-pentamethyldisilanyl)pyridine, analogous to compounds **1** and **2** were unsuccessful [24]. The reactions of 2-bromo-3-(1,1,2,2,2-pentamethyldisilanyl)pyridine with 1-octyne, 1-ethynylcyclohexene and ethynylcyclohexane did not afford 2-ethynyl-3-(1,1,2,2,2-pentamethyldisilanyl)pyridine derivatives. Many products were also detected in the reaction mixture by GLC and GPC.

Figure 4 illustrates a possible mechanistic interpretation of the reaction course. Due to the steric hindrance, the [RhCl(CO)_2_]_2_ binuclear complex decomposed into monomer complex RhCl(CO)_x_. A neutral RhCl(CO)_x_ is the real catalytic species. The coordination of RhCl(CO)_x_ to compound **1** resulted in model **0**, where the Si-Si bond was only slightly stretched from **1** (2.393 Å from 2.389 Å when x = 1). Therefore, model **0** is not shown in Figure 4, but was included in the DFT calculations, as shown in Appendix A. By way of **TS 0**–**1**, the Si-Si bond was cleaved and **Model 1** was formed. From **Model 1**, a trimethylsilyl group on the rhodium atom migrated to the carbon atom of the C=C bond (**Model 2**) via **TS 1–2**. The final change was a crossing of Me_3_Si and RhCl(CO)_x_ groups over the C=C bond via **TS 2–3**. This structure took a skew configuration, and counted the highest energy along the reaction coordinate (when x = 1). **Model 3** was generated from **TS 2–3**, and the elimination of rhodium species from **Model 3** afforded compound **3** (**Model 3** is not shown in Figure 4, but is shown in the Appendix A).

### 2.2. Theoretical Study

DFT calculations were conducted for the reaction mechanisms from compound **1** to compound **3** (*trans*-bis-silylation product), and also for the corresponding *cis*-bis-silylation product, 8-[bis(trimethylsilyl)methylene]-7,7-dimethyl-2-aza-7-silabicyclo [4.2.0]octa-1,3,5-triene, for comparison. The Gaussian09 program package [26] was employed along with the Becke’s three-parameter Lee–Yang–Parr hybrid functional [27]. Los Alamos effective core potentials [28] and the Dunning/Huzinaga full double basis sets [29] were used for the Rh atom. The 6-311G(d) basis sets were used for H, C, N, O, Si, and Cl atoms.

Firstly, transition states (TSs) were searched based on Figure 4. Then, for each TS, the intrinsic reaction coordinate (IRC) [30] was evaluated for both directions (reactant and product). At the end of IRC, normal optimization was followed until the two local minima (LMs) were reached.

Two catalyst models were used: RhCl(CO)_2_ and RhCl(CO) in the DFT calculations. The former is formed by the decomposition of [RhCl(CO)_2_]_2_, and the latter is formed by the disproportionation of the former, i.e., 2RhCl(CO)_2_ → RhCl(CO) + RhCl(CO)_3_.

**Model 0** is a combined system of compound **1** and RhCl(CO)_x_. The reaction proceeded in the order of **Model 0** → **TS 0–1** → **Model 1** → **TS 1–2** → **Model 2** → **TS 2–3** → **Model 3** (compound **3** + RhCl(CO)_x_). **Model 3** corresponds to compound **3** with Rh complex. All the optimized structures for LMs and TSs on the trans route are shown in the Appendix A. The energy change, along the reaction coordinate, are shown in Figure 1.

For comparison, the reaction mechanisms from compound **1** to compound **3b**, 8-[bis(trimethylsilyl)methylene]-7,7-dimethyl-2-aza-7-silabicyclo [4.2.0]octa-1,3,5-triene (*cis*-bis-silylation product), were also investigated based on Appendix A. All the LMs and TSs on the cis route are shown in the Appendix A. The energy change, along the reaction coordinate, are shown in Figure 2.

Figure 1 and Figure 2 show the SCF energy and the free energy, referring to the sum of energies of [RhCl(CO)_2_]_2_ and compound **1**, where the destabilization due to decomposition or disproportionation was taken into consideration. Both of the energies changed in parallel, and the location of the rate determining step did not depend on the two criteria.

Adopting the free energy, the rate determining steps were **TS 2–3** and **TS 2–3B** for the trans and cis routes, and their activation energies were 169 and 189 kJ mol^−1^, respectively, with the RhCl(CO) model. With the RhCl(CO)_2_ model, the rate determining step for the cis route was **TS 2–3B** with an activation energy of 170 kJ mol^−1^; however, the rate determining step for the trans route was **TS 0–1** with an activation energy of 123 kJ mol^−1^. Thus, both of the calculation models afforded the same result: that the *trans*-bis-silylation is the more stable product with respect to the activation energy, in accordance with our experimental result. Although the reactions with the RhCl(CO) model seem to proceed on the lower potential energy surfaces, the RhCl(CO)_2_ mode is a more promising catalyst for two reasons. (1) The activation energies for the rate determining step were 169 and 123 kJ mol^−1^ with RhCl(CO) and RhCl(CO)_2_ modes for the trans route. (2) The destabilization by disproportionation ([RhCl(CO)_2_]_2_ → RhCl(CO) + RhCl(CO)_3_) was 117 and 59 kJ mol^−1^ for the SCF and free energies. On the other hand, the decomposition energies ([RhCl(CO)_2_]_2_ → 2RhCl(CO)_2_) were 30 and −36 kJ mol^−1^, respectively. At the free energy level, the decomposition became stabilization. The present work is the first to investigate the two catalyst models intensively.

## 3. Conclusions

We have described here the rhodium-catalyzed reactions of 2-ethynyl-3-(1,1,2,2,2-pentamethyldisilanyl)pyridine derivatives. The reactions of 3-(1,1,2,2,2-pentamethyldisilanyl)-2-(trimethylsilylethynyl)pyridine (**1**) in the presence of a catalytic amount of rhodium complexes proceeded to give the pyridine-fused silole, 1,1-dimethyl-2,3-bis(trimethylsilyl)-1*H*-silolo(3,2-*b*)pyridine (**3**). Similar treatment of 2-ethynyl-3-(1,1,2,2,2-pentamethyldisilanyl)pyridine (**2**) afforded 1,1-dimethyl-3-(trimethylsilyl)-1*H*-silolo(3,2-*b*)pyridine (**4**). DFT calculations were performed to rationalize the formation of compound **3** via the intramolecular *trans*-bis-silylation of compound **1**. The synthesis of 2-(alkyl-substituted ethynyl)-3-(1,1,2,2,2-pentamethyldisilanyl)pyridine was unsuccessful.

## 4. Materials and Methods

### 4.1. General Procedure

All reactions were carried out under an inert atmosphere using dry nitrogen. NMR spectra were recorded on a JMN–ECS400 spectrometer using a deuteriochloroform solution. Low-resolution mass spectrometry was performed on a JEOL JMS-700 mass spectrometer. High-resolution mass spectrometry (HR-MS) was performed on a JEOL JMS-700 mass spectrometer. Gas chromatographic separations were carried out using a column (3 m × 10 mm) packed with 30% silicone on chromosorb W AM DMCS 80/100. Gel permeation chromatographic analysis was performed with a Model LC-908 Recycling Preparative HPLC (Japan Analytical Industry Co., Ltd., Tokyo, Japan). Column chromatography was performed using a silica gel column (Wakogel C–300; Wako Pure Chemical Industries, Osaka, Japan). Bis(triphenylphosphine)palladium(II) dichloride [PdCl_2_(PPh_3_)_2_], copper(I) iodide (CuI), Di-μ-chloro-tetracarbonyldirhodium(I) [RhCl(CO)_2_], Bis(norbornadiene-μ-chlororhodium) [RhCl(nbd)]_2_, and tris(triphenylphosphine)rhodium(I) chloride [RhCl(PPh_3_)_3_] were purchased from Sigma-Aldrich, St. Louis, MO, USA. Potassium carbonate, triethylamine, methanol, and toluene were purchased from Tokyo Kasei Kogyo, Tokyo, Japan. Triethylamine was distilled over potassium hydroxide under nitrogen just before use, and toluene was distilled from sodium benzophenone ketyl under nitrogen just before use. 2-Bromo-3-(1,1,2,2,2-pentamethyldisilanyl)pyridine was prepared as reported in the literature [24].

### 4.2. Procedures

Synthesis of Compound **1**. In a 300-mL three-necked flask fitted with a stirrer, reflux condenser, and dropping funnel, 2-bromo-3-(1,1,2,2,2-pentamethyldisilanyl)pyridine (6.108 g, 21.2 mmol), bis(triphenylphosphine)palladium(II) dichloride (0.760 g, 1.08 mmol), and copper(I) iodide (0.214 g, 1.12 mmol) were added to 50 mL of dry triethylamine. To this mixture, ethynyltrimethylsilane (2.553 g, 26.0 mmol) was added dropwise at room temperature. The mixture was heated to reflux for 12 h. The solution was then hydrolyzed, and the organic layer was separated, washed with water, and dried over anhydrous magnesium sulfate. The solvent was then evaporated, and the residue was chromatographed on a silica gel column and eluted with hexane-ethyl acetate (10:1) to obtain 1.571 g (24% yield) of compound **1**: ^1^H NMR δ(CDCl_3_) 0.09 (s, 9H, Me_3_Si), 0.27 (s, 9H, Me_3_Si), 0.45 (s, 6H, Me_2_Si), 7.17 (dd, 1H, pyridyl-ring proton, J = 7.6, 5.2 Hz), 7.70 (dd, 1H, pyridyl-ring proton, J = 7.6, 2.0 Hz), 8.51 (dd, 1H, pyridyl-ring proton, J = 5.2, 2.0 Hz). All spectral data for compound **1** were identical to those of an authentic sample [24].

Synthesis of Compound **2**. A mixture of compound **1** (1.571 g, 5.14 mmol), potassium carbonate (0.982 g, 7.11 mmol), and methanol (50 mL) was stirred at room temperature for 2 h. The reaction mixture was concentrated under reduced pressure, and hexane (15 mL) and water (30 mL) were added to the residue. The layers were separated and the aqueous layer was extracted with hexane (4 × 15 mL). The solvent was then evaporated, and the residue was chromatographed on a silica gel column, eluted with hexane-ethyl acetate (10:1), to obtain 0.995 g (83% yield) of compound **2**: HR-MS: calcd. for C_12_H_19_NSi_2_ (M^+^): 233.1056, found: 233.1052. MS *m*/*z* 233 (M^+^); ^1^H NMR δ(CDCl_3_) 0.10 (s, 9H, Me_3_Si), 0.44 (s, 6H, Me_2_Si), 3.24 (s, 1H, HC), 7.21 (dd, 1H, pyridyl ring proton, J = 7.8 Hz, 4.8 Hz), 7.72 (dd, 1H, pyridyl ring proton, J = 7.8 Hz, 2.0 Hz), 8.52 (dd, 1H, pyridyl ring proton, J = 4.8 Hz, 2.0 Hz); ^13^C NMR δ(CDCl_3_) −3.7 (Me_2_Si), 1.3 (Me_3_Si), 79.5, 85.0 (sp carbons), 122.8, 138.1, 142.0, 146.7, 149.5 (pyridyl ring and olefinic carbons); ^29^Si NMR δ(CDCl_3_) −20.7, −17.2.

Reaction of Compound **1** in the Presence of [RhCl(CO)_2_]_2_ Catalyst. In a 30 mL two-necked flask fitted with a reflux condenser were placed 0.375 g (1.23 mmol) of compound **1** and 0.047 g (0.121 mmol) of [RhCl(CO)_2_]_2_ in 5 mL of dry toluene. The mixture was heated to reflux for 12 h. The solution was then hydrolyzed, and the organic layer was separated, washed with water, and dried over anhydrous magnesium sulfate. The solvent was then evaporated, and the residue was chromatographed on a silica gel column, eluting with hexane-ethyl acetate (10:1), to obtain 0.172 g (46% yield) of compound **3**: HR-MS: calcd. for C_15_H_27_NSi_3_ (M^+^): 305.1451, found: 305.1455. MS *m*/*z* 305 (M^+^); ^1^H NMR δ(CDCl_3_) 0.29 (s, 9H, Me_3_Si), 0.34 (s, 6H, Me_2_Si), 0.38 (s, 9H, Me_3_Si), 6.98 (dd, 1H, pyridyl ring proton, J = 7.2 Hz, 5.2 Hz), 7.73 (dd, 1H, pyridyl ring proton, J = 7.2 Hz, 1.6 Hz), 8.46 (dd, 1H, pyridyl ring proton, J = 5.2 Hz, 1.6 Hz); ^13^C NMR δ(CDCl_3_) −2.7 (Me_2_Si), 2.5, 2.7 (Me_3_Si), 120.5, 132.6, 138.1, 149.0, 165.0, 172.0, 176.6 (pyridyl ring and olefinic carbons); ^29^Si NMR δ(CDCl_3_) −9.8, −6.6, 10.1.

Reaction of Compound **1** in the Presence of [RhCl(nbd)]_2_ Catalyst. In a 30 mL two-necked flask fitted with a reflux condenser were placed 0.300 g (0.98 mmol) of compound **1** and 0.045 g (0.098 mmol) of [RhCl(nbd)]_2_ in 5 mL of dry toluene. The mixture was heated to reflux for 12 h. The solution was then hydrolyzed, and the organic layer was separated, washed with water, and dried over anhydrous magnesium sulfate. The solvent was then evaporated and the residue was chromatographed on a silica gel column, eluting with hexane-ethyl acetate (10:1), to obtain 0.018 g (6% yield) of compound **3**. All spectral data for compound **3** were identical to those of an authentic sample.

Reaction of Compound **1** in the Presence of RhCl(PPh_3_)_3_ Catalyst. In a 30 mL two-necked flask fitted with a reflux condenser were placed 0.575 g (1.88 mmol) of **1** and 0.176 g (0.190 mmol) of RhCl(PPh_3_)_3_ in 5 mL of dry toluene. The mixture was heated to reflux for 12 h. The solution was then hydrolyzed, and the organic layer was separated, washed with water, and dried over anhydrous magnesium sulfate. The solvent was then evaporated and the residue was chromatographed on a silica gel column, eluting with hexane-ethyl acetate (10:1), to obtain 0.030 g (5% yield) of compound **3**. The starting compound **1** was almost recovered (0.502 g). All spectral data for compound **3** were identical to those of an authentic sample.

Reaction of Compound **2** in the Presence of [RhCl(CO)_2_]_2_ Catalyst. In a 30 mL two-necked flask fitted with a reflux condenser were placed 0.134 g (0.574 mmol) of compound **1** and 0.023 g (0.059 mmol) of [RhCl(CO)_2_]_2_ in 5 mL of dry toluene. The mixture was heated to reflux for 12 h. The solution was then hydrolyzed, and the organic layer was separated, washed with water, and dried over anhydrous magnesium sulfate. The solvent was then evaporated and the residue was chromatographed on a silica gel column, eluting with hexane-ethyl acetate (10:1), to obtain 0.064 g (48% yield) of compound **4**: HR-MS: calcd. for C_12_H_19_NSi_2_ (M^+^): 233.1056, found: 233.1060. MS *m*/*z* 233 (M^+^); ^1^H NMR δ(CDCl_3_) 0.30 (s, 9H, Me_3_Si), 0.31 (s, 6H, Me_2_Si), 6.99 (s, 1H, olefinic proton), 7.00 (dd, 1H, pyridyl ring proton, J = 7.0 Hz, 5.2 Hz), 7.75 (dd, 1H, pyridyl ring proton, J = 7.0 Hz, 2.0 Hz), 8.48 (dd, 1H, pyridyl ring proton, J = 5.2 Hz, 2.0 Hz); ^13^C NMR δ(CDCl_3_) −4.3 (Me_2_Si), −1.0 (Me_3_Si), 120.6, 132.3, 138.7, 148.4, 149.6, 168.7, 170.7 (pyridyl ring and olefinic carbons); ^29^Si NMR δ(CDCl_3_) −6.4, 2.9.

Reaction of Compound **2** in the Presence of [RhCl(nbd)]_2_ Catalyst. In a 30 mL two-necked flask fitted with a reflux condenser were placed 0.130 g (0.557 mmol) of compound **1** and 0.025 g (0.054 mmol) of [RhCl(nbd)]_2_ in 5 mL of dry toluene. The mixture was heated to reflux for 12 h. The solution was then hydrolyzed, and the organic layer was separated, washed with water, and dried over anhydrous magnesium sulfate. The solvent was then evaporated and the residue was chromatographed on a silica gel column, eluting with hexane-ethyl acetate (10:1), to obtain 0.007 g (5% yield) of compound **4**. All spectral data for compound **4** were identical to those of an authentic sample.

Reaction of Compound **2** in the Presence of RhCl(PPh_3_)_3_ Catalyst. In a 30 mL two-necked flask fitted with a reflux condenser were placed 0.094 g (0.403 mmol) of compound **2** and 0.037 g (0.040 mmol) of RhCl(PPh_3_)_3_ in 5 mL of dry toluene. The mixture was heated to reflux for 12 h. The solution was then hydrolyzed, and the organic layer was separated, washed with water, and dried over anhydrous magnesium sulfate. The solvent was then evaporated and the residue was chromatographed on a silica gel column, eluting with hexane-ethyl acetate (10:1). The starting compound **2** was recovered (0.075 g).

Reaction of 2-Bromo-3-(1,1,2,2,2-pentamethyldisilanyl)pyridine with 1.9 equivalent of 3-Phenyl-1-propyne in the Presence of Palladium and Copper Catalysts. In a 300 mL two-necked flask fitted with a reflux condenser, 2-bromo-3-(1,1,2,2,2-pentamethyldisilanyl)pyridine (2.675 g, 9.28 mmol), bis(triphenylphosphine)palladium(II) dichloride (0.316 g, 0.450 mmol), and copper(I) iodide (0.086 g, 0.452 mmol) were added to 50 mL of dry triethylamine. To this mixture, 3-phenyl-1-propyne (2.091 g, 18.0 mmol) was added dropwise at room temperature, after which the mixture was heated to reflux for 12 h. The solution was then hydrolyzed, and the organic layer was separated, washed with water, and dried over anhydrous magnesium sulfate. The solvent was then evaporated, and the residue was chromatographed on a silica gel column, eluting with hexane-ethyl acetate (50:1) to obtain 0.933 g (23% yield) of compound 6: HR-MS: calcd. for C_28_H_33_NSi_2_ (M^+^): 439.2152, found: 439.2150. MS *m*/*z* 439 (M^+^); ^1^H NMR δ(CDCl_3_) 0.10 (s, 9H, Me_3_Si), 0.45 (s, 6H, Me_2_Si), 3.14-3.27 (m, 2H, methylene), 3.66 (s, 2H, methylene), 6.18 (br s, 1H, olefinic proton), 7.12−7.35 (m, 11H, phenyl and pyridyl ring protons), 8.10 (s, 1H, pyridyl ring proton), 8.62 (br d, 1H, pyridyl ring proton, J = 3.2 Hz); ^13^C NMR δ(CDCl_3_) −0.9 (Me_3_Si), 1.0, 1.3 (MeSi), 26.3, 37.6 (CH_2_), 85.7, 106.4 (sp carbons), 120.0, 125.9, 126.6, 127.8, 128.3, 128.5, 128.7, 129.5, 136.0, 140.1, 140.6, 143.9, 147.5, 149.7, 165.8 (phenyl, pyridyl ring, and olefinic carbons); ^29^Si NMR δ(CDCl_3_) −22.2, −3.3.

Reaction of 2-Bromo-3-(1,1,2,2,2-pentamethyldisilanyl)pyridine with 0.96 equivalent of 3-Phenyl-1-propyne in the Presence of Palladium and Copper Catalysts. In a 300 mL two-necked flask fitted with a reflux condenser, 2-bromo-3-(1,1,2,2,2-pentamethyldisilanyl)pyridine (1.800 g, 6.24 mmol), bis(triphenylphosphine)palladium(II) dichloride (0.211 g, 0.301 mmol), and copper(I) iodide (0.057 g, 0.299 mmol) were added to 50 mL of dry triethylamine. To this mixture, 3-phenyl-1-propyne (0.695 g, 5.98 mmol) was added dropwise at room temperature, after which the mixture was heated to reflux for 12 h. The solution was then hydrolyzed, and the organic layer was separated, washed with water, and dried over anhydrous magnesium sulfate. The solvent was then evaporated, and the residue was chromatographed on a silica gel column eluting with hexane-ethyl acetate (50:1) to obtain 0.334 g (12% yield based on 2-bromo-3-(1,1,2,2,2-pentamethyldisilanyl)pyridine) of compound **6**. All spectral data for compound 6 were identical to those of an authentic sample.

Reaction of 2-Bromo-3-(1,1,2,2,2-pentamethyldisilanyl)pyridine with 1-Hexyne in the Presence of Palladium and Copper Catalysts. In a 300 mL two-necked flask fitted with a reflux condenser, 2-bromo-3-(1,1,2,2,2-pentamethyldisilanyl)pyridine (5.474 g, 19.0 mmol), bis(triphenylphosphine)palladium(II) dichloride (0.667 g, 0.950 mmol), and copper(I) iodide (0.181 g, 0.950 mmol) were added to 50 mL of dry triethylamine. To this mixture, 1-hexyne (3.194 g, 38.9 mmol) was added dropwise at room temperature, after which the mixture was heated to reflux for 12 h. Many products were detected in the reaction mixture by GLC and GPC. The solution was then hydrolyzed, and the organic layer was separated, washed with water, and dried over anhydrous magnesium sulfate. The solvent was then evaporated, and the residue was chromatographed on a silica gel column, eluting with hexane-ethyl acetate (10:1). Although the ^1^H NMR spectrum showed the existence of 2-(hex-1-yn-1-yl)-3-(1,1,2,2,2-pentamethyldisilanyl)pyridine produced from a Sonogashira coupling reaction of 2-bromo-3-(1,1,2,2,2-pentamethyldisilanyl)pyridine and 1-hexyne, analogous to compounds **1** and **2**, all attempts to isolate the compound were unsuccessful.

Reaction of 2-Bromo-3-(1,1,2,2,2-pentamethyldisilanyl)pyridine with 1-Octyne in the Presence of Palladium and Copper Catalysts. In a 300 mL two-necked flask fitted with a reflux condenser, 2-bromo-3-(1,1,2,2,2-pentamethyldisilanyl)pyridine (4.285 g, 14.9 mmol), bis(triphenylphosphine)palladium(II) dichloride (0.529 g, 0.754 mmol), and copper(I) iodide (0.144 g, 0.756 mmol) were added to 50 mL of dry triethylamine. To this mixture, 1-octyne (3.270 g, 29.7 mmol) was added dropwise at room temperature, after which the mixture was heated to reflux for 12 h. Many products were detected in the reaction mixture by GLC and GPC. The solution was then hydrolyzed, and the organic layer was separated, washed with water, and dried over anhydrous magnesium sulfate. The solvent was then evaporated, and the residue was chromatographed on a silica gel column, eluting with hexane-ethyl acetate (10:1). No 2-(oct-1-yn-1-yl)-3-(1,1,2,2,2-pentamethyldisilaneyl)pyridine would be produced from a Sonogashira coupling reaction of 2-bromo-3-(1,1,2,2,2-pentamethyldisilanyl)pyridine, and 1-octyne, analogous to compounds **1** and **2**, was not detected.

Reaction of 2-Bromo-3-(1,1,2,2,2-pentamethyldisilanyl)pyridine with 1-Ethynylcyclohexene in the Presence of Palladium and Copper Catalysts. In a 300 mL two-necked flask fitted with a reflux condenser, 2-bromo-3-(1,1,2,2,2-pentamethyldisilanyl)pyridine (1.899 g, 6.59 mmol), bis(triphenylphosphine)palladium(II) dichloride (0.667 g, 0.349 mmol), and copper(I) iodide (0.068 g, 0.357 mmol) were added to 50 mL of dry triethylamine. To this mixture, 1-ethynylcyclohexene (1.513 g, 14.3 mmol) was added dropwise at room temperature, after which the mixture was heated to reflux for 12 h. Many products were detected in the reaction mixture by GLC and GPC. The solution was then hydrolyzed, and the organic layer was separated, washed with water, and dried over anhydrous magnesium sulfate. The solvent was then evaporated, and the residue was chromatographed on a silica gel column, eluting with hexane-ethyl acetate (10:1). No 2-(cyclohex-1-en-1-ylethynyl)-3-(1,1,2,2,2-pentamethyldisilanyl)pyridine would be produced from a Sonogashira coupling reaction of 2-bromo-3-(1,1,2,2,2-pentamethyldisilanyl)pyridine, and 1-ethynylcyclohexene, analogous to compounds **1** and **2**, was not detected.

Reaction of 2-Bromo-3-(1,1,2,2,2-pentamethyldisilanyl)pyridine with Ethynylcyclohexane in the Presence of Palladium and Copper Catalysts. In a 300 mL two-necked flask fitted with a reflux condenser, 2-bromo-3-(1,1,2,2,2-pentamethyldisilanyl)pyridine (1.558 g, 5.40 mmol), bis(triphenylphosphine)palladium(II) dichloride (0.175 g, 0.249 mmol), and copper(I) iodide (0.047 g, 0.247 mmol) were added to 50 mL of dry triethylamine. To this mixture, ethynylcyclohaxane (1.064 g, 9.84 mmol) was added dropwise at room temperature, after which the mixture was heated to reflux for 12 h. Many products were detected in the reaction mixture by GLC and GPC. The solution was then hydrolyzed, and the organic layer was separated, washed with water, and dried over anhydrous magnesium sulfate. The solvent was then evaporated, and the residue was chromatographed on a silica gel column, eluting with hexane-ethyl acetate (10:1). No 2-(cyclohexylethynyl)-3-(1,1,2,2,2-pentamethyldisilanyl)pyridine would be produced from a Sonogashira coupling reaction of 2-bromo-3-(1,1,2,2,2-pentamethyldisilanyl)pyridine, and ethynylcyclohexane, analogous to compounds **1** and **2**, was not detected.

## Data Availability

The data presented in this study are available in the article and Appendix A.

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
