# Peer review of "Rhodium-Catalyzed Trans-Bis-Silylation Reactions of 2-Ethynyl-3-pentamethyldisilanylpyridines"

_molecules, 2023, doi:10.3390/molecules28083284_

Round 1

Reviewer 1 Report

The ms of Kobayashi and co-worker is sound. Already the preparation of the starting material is interesting. This referee misses, in respect of the preparation of compounds 3 and 4, the phrase "cyclization" from the abstract. The preparation of 6 was also forgotten to be mentioned in the abstract, so pls try to make this important part more informative.

Re the Experimental, compounds 1 and 3 were already described earlier. This referee thinks taht it would be better to give some spectral data (e.g. 1H NMR) for these species, compared with those of the literature data. So two lists would be desirable.

What really is a shortcoming that there is only a few compounds prepared. However, this might be compensated by the new methods developed.

Reviewer 2 Report

This manuscript describes the Rh–catalyzed reactions of 2-ethynyl-3-pentamethyldisilanylpyridine derivatives and DFT calculations to investigate the energy and structural changes in the synthesis route from 3- (1,1,2,2,2-pentamethyldisilanyl)-2-(trimethylsilylethynyl)pyridine 1 to trans-bis-silylation product 3. I went through the literature regarding trans-bis-silylation reactions of alkynes and I did not find much significant novelty from this manuscript, way of presentation, as well as yields of compounds, are very low. In my opinion, the scientific significance of this work is very low and not fit for "molecules" criteria and I am sorry to have to reject this paper might be suitable for less impact journals would be more appropriate like heterocyclic chemistry journals. 

Reviewer 3 Report

In this article, Naka and Kobayashi report the catalytic trans-bis-silylation of alkynyl-disilane substituted pyridines mediated by rhodium complexes. The process itself is interesting from a synthetic point of view and, apparently, trans-bis-silylation is rather uncommon. From this perspective, my feeling is that these results are worthy of publication but the paper needs major revisions both from the English grammar and for the way the results are presented.  First of all, I find a little bit confusing the way the synthesis of the alkynyl-substituted pyridines has been shown. The authors start by explaining the systems that actually work (compounds 1 and 3), then moving to catalysis and afterwards coming back to the attempted synthesis of other alkynyl-disila-substituted pyridines. One of these system was not possible to be prepared due to further reaction with a second equivalent of the alkyne that is unavoidable (compound 6), whereas other alkynes didn’t undergo the Sonogashira coupling. I suggest the authors to group all this chemistry before explaining the catalytic processes, indicating that this is a drawback of the protocol. Once the starting materials 1 and 2 are ready, several catalysts have been tested. I believe that the inclusion of a table with the catalysts used and the reaction conditions will be clearly beneficial for the reader. Regarding the DFT calculations, I find difficult to follow the discussion without a diagram with all intermediates and TS with their corresponding energies (on the other hand, are these energies DG or DH?, the author should give DG values considering that entropic factors might be relevant). One important thing that needs revision is that the authors have used as the energy reference the sum of energies of [RhCl(CO)2]2 and RhCl(CO). However, production of the highly unsaturated, 12-electron RhCl(CO) complex from the starting rhodium-dimer might be highly energy demanding and this should be shown in the energy diagrams. In this sense, have the authors considered other possible reaction pathways, as for example breaking the dimer into its monomers followed by coordination of the alkyne and then oxidative addition of the Si-Si bond leading to an octahedral Rh(III) complex without the need of liberation of CO? If CO is released, the reaction in the presence of CO in excess should be inhibited, at least partially. Overall, I suggest that the authors address the deficiencies since my feeling is that the article is too preliminary for publication at this point.

Some typos:

Page 1, line 26: “has been reported…” should be “have been reported…”

Page 2, line 73:  “ a signal at….” Should be “signals at….”

Page 3, line 93: “product 6 produced…” should be “ product 6 was produced”

Page 3, line 111: “afforded no…” should be “didn’t afford”

Page 4, lines 118-120: change Ru by Rh

Page 5, line 141: Seached should be searched

Page 6, line 145: compound should be compound

Page 6, figure 1 caption: “Energy diagrams for from model” should be “Energy diagrams for model”

Page 7, line 224; page 8 lines 237, 245, 254, 267 and 275: “reflex” should be “reflux”

Round 2

Reviewer 2 Report

The authors responded to some of the comments from other referees and I leave it to the editors' decision.

Author Response

We thank the reviewer for his/her comments. 

Reviewer 3 Report

In the new version, the authors have answered most of my concerns. Still, I believe that a reorganization of the synthesis of precursors and catalytic studies should be done. The consideration of complex [RhCl(CO)2] as a possible catalytic intermediate has been now considered, offering an alternative route to the production of [RhCl(CO)]. The new calculations predict that [RhCl(CO)2] should be more favorable, since the global energy is as low as 123 kJmol-1, consistent with the reaction conditions used in the real system (110 oC in toluene). For the system [RhCl(CO)], although it yields more stable potential energies, the fact that 169 kJ mol-1 are required as a consequence of formation of a very stable intermediate (model 1), my feeling is that this energy barriers in the subsequent steps are exceedingly high to be overcome under the experimenatl conditions (46 kJmol-1 - or nearly 11 kcal mol-1 -, higher than those considering complex [RhCl(CO)2]), and therefore I believe that [RhCl(CO)2] is more likely involved in the catalysis. Thus, I suggest the authors to rewrite the conclusions to take into account these numbers. 

Author Response

We thank the reviewer for his/her comments.   On line 263-271, we added “Although the reactions with RhCl(CO) model seem to proceed on the lower potential energy surfaces, RhCl(CO)2 mode is more promising catalyst by the two reasons.  (1) The activation energies for the rate determining step are 169 and 123 kJ mol-1 with RhCl(CO) and RhCl(CO)2 modes for the trans route.  (2) The destabilization by disproportionation ([RhCl(CO)2]2 → RhCl(CO) + RhCl(CO)3) is 117 and 59 kJ mol-1 for the SCF and free energies.  On the other hand, the decomposition energies ([RhCl(CO)2]2 → 2RhCl(CO)2) are 30 and –36 kJ mol-1, respectively.   At the free energy level, the decomposition becomes stabilization.  The present work is the first one which investigated the two catalyst models intensively.” 

On line 271, we removed “Furthermore, judging from the relative energy height in the figures, the reactions with RhCl(CO) model proceed on the more stable potential energy surfaces.  This result is also consistent with the fact that the RhCl(CO) model was adopted in previous studies [31].” 

On line 566, we deleted “31. Yu, Z.-X.; Wender, P. A.; Houk, K. N. On the Mechanism of [Rh(CO)2Cl]2-Catalyzed Intermolecular (5 + 2) Reactions between Vinylcyclopropanes and Alkynes, J. Am. Chem. Soc. 2004, 126, 9154-9155.”